# Consistent RNA sequencing contamination in GTEx and other data sets

Tim O. Nieuwenhuis[1,2], Stephanie Y. Yang[2], Rohan X. Verma[1], Vamsee Pillalamarri[2], Dan E. Arking[2], Avi Z. Rosenberg[1], Matthew N. McCall[3] & Marc K. Halushka [1✉]

A challenge of next generation sequencing is read contamination. We use Genotype-Tissue Expression (GTEx) datasets and technical metadata along with RNA-seq datasets from other studies to understand factors that contribute to contamination. Here we report, of 48 analyzed tissues in GTEx, 26 have variant co-expression clusters of four highly expressed and pancreas-enriched genes (*PRSS1*, *PNLIP*, *CLPS*, and/or *CELA3A*). Fourteen additional highly expressed genes from other tissues also indicate contamination. Sample contamination is strongly associated with a sample being sequenced on the same day as a tissue that natively expresses those genes. Discrepant SNPs across four contaminating genes validate the contamination. Low-level contamination affects ~40% of samples and leads to numerous eQTL assignments in inappropriate tissues among these 18 genes. This type of contamination occurs widely, impacting bulk and single cell (scRNA-seq) data set analysis. In conclusion, highly expressed, tissue-enriched genes basally contaminate GTEx and other datasets impacting analyses.

---

[1] Department of Pathology, Johns Hopkins University SOM, Baltimore, MD 21205, USA. [2] McKusick-Nathans Institute, Department of Genetic Medicine, Johns Hopkins University SOM, Baltimore, MD 21205, USA. [3] Department of Biostatistics and Computational Biology, University of Rochester Medical Center, Rochester, NY 14642, USA. ✉email: mhalush1@jhmi.edu

The rise of next-generation sequencing has allowed for unparalleled data generation for a variety of nucleic acid studies including RNA expression. As cost per basepair decreases, more large-scale transcriptome projects can be performed that will inform on tissue expression patterns in health and disease[1–4]. These data sources are generally publicly available and have been used by hundreds of researchers for secondary analyses of high impact[5,6].

Limitations exist for all –omics technologies, including RNA sequencing (RNA-Seq). Issues of hybridization biases, library preparation biases, and computational biases such as positional fragment bias are known limitations of RNA-Seq experiments[7–9]. Another challenge of high-throughput RNA-Seq is contamination, leading to the presence of sequence data within a data set of one sample that originates from a separate sample. This contamination can come from many different aspects of the modern sequencing process, such as human error, machine or equipment contamination, intrinsic preparation and sequencing errors, and computational errors, including errors that can occur based on the multiplexing methods used (https://www.illumina.com/content/dam/illumina-marketing/documents/products/whitepapers/index-hopping-white-paper-770-2017-004.pdf, visited 02/24/2020)[10,11]. For single cell RNA-Seq (scRNA-Seq), doublets or multiplets (2+ cells partitioned together) can cause cross-contamination resulting in expression hybrids[12]. Compared with RNA-Seq, contamination has been better characterized for DNA sequencing projects[13–15].

The Genotype-Tissue Expression project (GTEx) aims to create a large publicly available database of tissue-specific expression quantitative trait loci (eQTL) from over 40 tissues[1]. It is an ongoing project with over 700 individuals and 11,000 tissue samples. GTEx identifies eQTLs by associating genotypes called from whole-genome sequencing with gene expression levels obtained from bulk RNA-Seq. GTEx has made their RNA-Seq, phenotype, genotype, and technical data available for public access with permission.

In an analysis of variation in the GTEx RNA-Seq data (V7), we detect unexpected sources of variation that we hypothesize are likely contaminating sequence reads found at low, but variable levels across different tissues. Herein, we describe how we identify the source of contamination and establish basal rates of contamination in the GTEx bulk RNA-Seq data. We further demonstrate the universality of highly expressed genes contaminating other samples.

## Results

**Extreme tissue variation identifies gene signature patterns**. We embarked on a project to expand our initial description of the causes of lung expression variation in GTEx to all tissue samples[16]. We used DESeq2 variance stabilizing transformation (VST) to normalize read counts from 11,340 samples[16,17]. Then we filtered genes in each tissue keeping those with a mean transformed count >5. The median number of genes above the expression threshold was 17,729 with the highest and lowest gene counts being 23,930 and 13,807 in the testis and whole blood, respectively. As previously described, we correlated and hierarchically clustered variable genes (more than four variance across samples) for all tissues with >70 samples (N = 48) in the GTEx data set V7[16]. Our algorithm identified multiple gene clusters per tissue, based on their Kendall's tau correlations (Fig. 1a). It additionally reported non-clustering, highly variable genes. Most clusters were the result of biologic and phenotypic features related to the tissues. For example, a cluster of Y chromosome genes and *XIST* appeared in 42 of 43 non-sex specific tissues. However, there was one consistent cluster of 3–4 genes

(*PNLIP, PRSS1, CELA3A,* and/or *CLPS*) identified in 26 of the 48 tissues that did not have an intuitive biological explanation. These genes are highly expressed and pancreas acinar-cell specific[18]. To identify other highly expressed tissue-enriched genes appearing variably in other samples, we cross-referenced a list of tissue-enriched proteins generated by the Human Protein Atlas (HPA) to the GTEx transcripts per million (TPM) data (Table 1)[19,20]. We noted 18 genes from seven tissues including two esophagus genes *KRT13* and *KRT4* that are highly expressed in their native tissue and identified as variable in five or more other unrelated tissues (Fig. 1a, Supplementary Fig. 1).

As both abundant and tissue-enriched genes were unlikely to be randomly and lowly expressed in a range of other tissues, we performed analyses to determine the source of the contamination.

We first questioned if the contamination occurred during tissue harvesting, hypothesizing that occasionally small fragments of a tissue could contaminate a separate sample from shared dissection tools or surfaces. A pancreas gene contamination cluster was found in transformed fibroblasts, which were grown over multiple passages and would not retain other cell types over that time period, challenging this as a source of contamination (Supplementary Fig. 1).

**Sequencing date is the main source of contamination**. We then queried if other technical sources of contamination may be related. Other GTEx metadata available included nucleic acid isolation date and sequencing date, which were both interrogated for associations with this contamination. The normalized score of four pancreas-enriched genes (*PRSS1, CLPS, PNLIP,* and *CELA3A*) was significantly higher in non-pancreas samples if they were sequenced on the same day as a pancreas sample (Wilcox rank sum test, $p < 5e\text{-}324$, Fig. 1b, c). We then performed linear mixed model analyses across non-contaminating tissues to understand expression levels of genes known to be highly enriched and highly expressed in the pancreas or esophagus, but were found lowly and variably in other tissues (ex. coronary artery, liver, tibial nerve). Because the majority of samples were sequenced on the same day as a pancreas or esophagus–mucosa sample, for model robustness, we limited our models to only tissues with >40 samples not sequenced on the same data as a respective contaminating tissue (Supplementary Data 1 and 2).

After adjusting for tissue type, the pancreas contamination score (average of normalized expression values for *PRSS1, CLPS, PNLIP,* and *CELA3A*) was on average 0.175 higher if the tissue was isolated on the same day as a pancreas sample ($p = 3.9e\text{-}12$, linear regression), but on average 0.863 higher if the tissue was sequenced on the same day as a pancreas sample ($p = 9.5e\text{-}237$, linear regression) (Supplementary Data 3).

A similar linear mixed model was used to evaluate the esophagus contamination score (average of normalized expression values for two highly expressed esophagus-enriched genes *KRT4* and *KRT13*). The same model as above showed only sequencing a tissue on the same day as a esophagus sample to be significant ($p = 5e\text{-}260$) (Supplementary Data 4).

Despite the significance of sequencing date, some high scores came from samples that were not sequenced on the same days as pancreata. Focusing on just one these genes, *PRSS1*, it was clear that all of these samples were sequenced within a few days of a pancreas (Fig. 1d). This additionally implicated the library preparation process (for which date information is lacking in GTEx) as a potential source for contamination, as it is temporally related, but not identical, to sequencing date.

With this understanding of the temporal importance of sequencing date to gene expression, we then revisited the four genes making up the pancreas normalized score gene cluster. For

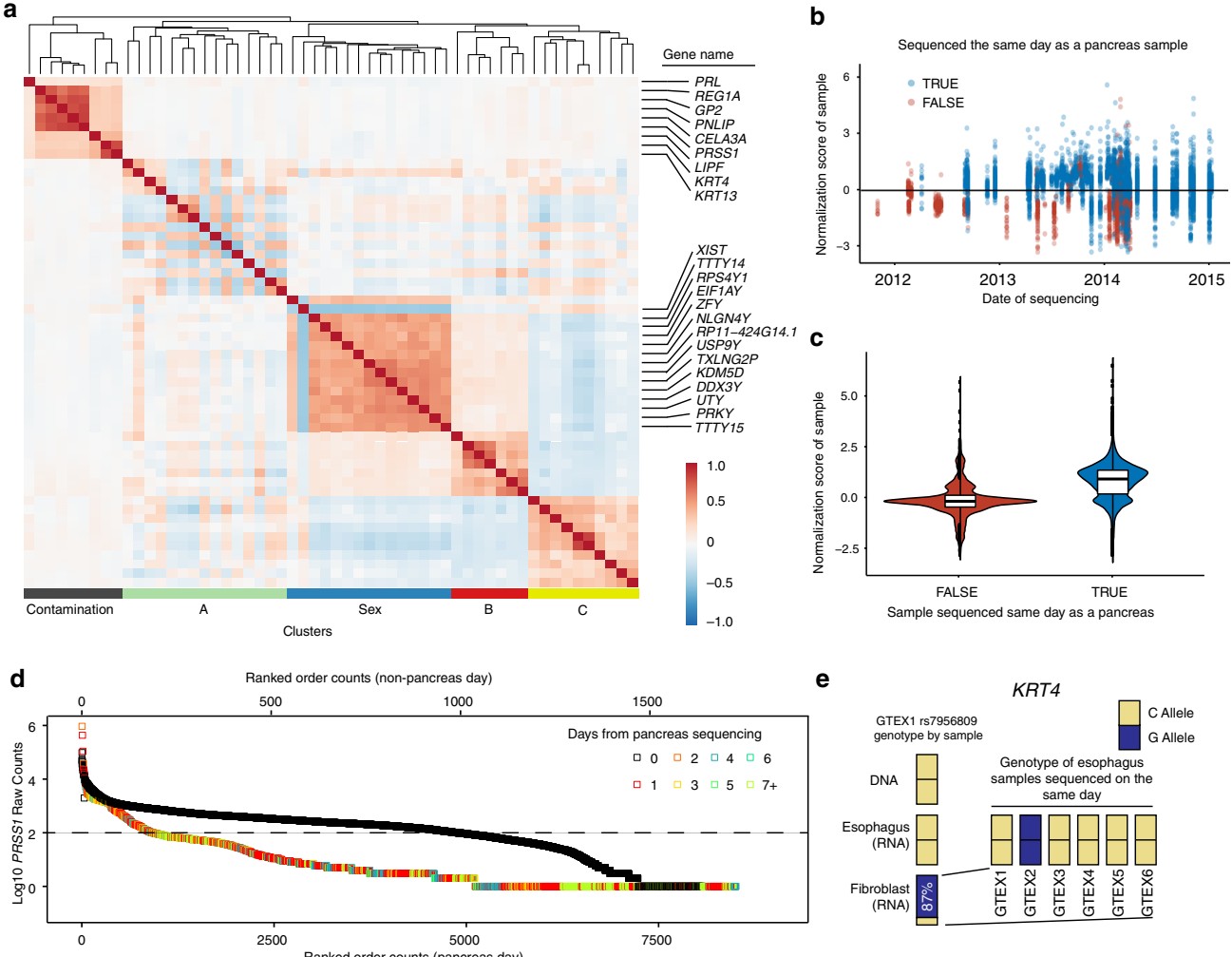

**Fig. 1 Identification and explanation of sequencing contamination. a** A correlation heatmap of highly variable subcutaneous adipose tissue genes across 442 subjects. Blue to red scale shows Kendall's tau correlation from −1 to 1. The genes within the contamination cluster and the sex cluster are given. The meaning of cluster A is unknown. Cluster B may relate to the percentage of smooth muscle cells and cluster C includes acute phase reactants. **b** Contamination normalized score values for non-pancreas tissue samples ($N = 11,366$) colored relative to being sequenced on the same day as a pancreas tissue. The solid black line denotes a $Z$ score of 0. **c** Violin plot of the same data showing a strong, but not complete correlation of sequencing on a pancreas day. The solid line in all boxplots represents the median of the data, whereas the lower and upper hinges correspond to the 25th percentile and 75th percentile, respectively. The whiskers represent the interquartile range × 1.5, and any outliers beyond the whiskers are represented as dots. **d** Ranked order of all samples either sequenced on the same day as a pancreas sample (black) or on a non-pancreas sequencing day (colors) for *PRSS1* read counts in log10. Among samples not sequenced on a pancreas day, 91% of samples with >100 reads were sequenced within 4 days of a known sequenced pancreas. The dashed line represents 100 reads. **e** Keratin 4 (*KRT4*) contaminating reads in GTEX-1's fibroblast RNA-Seq appear to have originated from GTEX2 esophagus mucosa tissue. By DNA and RNA of the appropriate tissue source of *KRT4*, sample GTEX-1 is homozygous for the C allele at rs7956809. The fibroblast sample is 87% G reads, primarily matching sample GTEX2. The read count depth at the SNP in the GTEX-1 esophagus was 85,803 and 204 for the GTEX-1 fibroblast.

all tissues, median normalized scores were higher on pancreas sequencing dates, and the division between the 26 tissues with clustering vs. the 21 without was a result of the high threshold for VST normalized counts we set for our pipeline (Supplementary Fig. 2).

**Genetic polymorphisms confirm contamination patterns**. To prove that pancreas/esophagus transcripts were contaminating from other (non-self) samples we identified incongruencies between a person's genotype (from DNA data) and the genotype in matching loci in the pancreas/esophagus contaminated RNA-Seq samples. We required both the individuals' DNA genotype and RNA-Seq files from contaminating tissues, in order to account for both RNA editing and preferential allele expression.

Based on these sample requirements and limited by available raw sequencing files, we identified 11 contaminated tissues to evaluate. For each, we obtained and processed their raw RNA-Seq FASTQ sequences to identify nucleotide variants in both their contaminated tissues and their matched pancreas or esophagus tissue (depending on the gene source of contamination). In addition, we used the GTEx filtered variant call format (VCF) file from the individual's sequenced DNA to further establish their SNP allele patterns. Across all tissues, 533 SNPs, rare variants, and private variants, were investigated in pancreas associated gene coding sequences (*PNLIP*, *CLPS*, and *CELA3A*) and 190 in esophagus associated gene coding sequences (*KRT13*, *KRT4*). As a comparison group, 287 variants were investigated in two control gene coding sequences (*GAPDH* and *RAB7A*) that have near

**Table 1 Eighteen highly expressed genes that appear to contaminate five or more GTEx tissues.**

| Gene | Times identified as variable in other tissues | Highest expressed GTEx tissue | GTEx TPM | Tissue* | GTEx TPM* | Independent study TPM* |
|---|---|---|---|---|---|---|
| *PRSS1* | 43 | Pancreas | 99,100 | Stomach | 28.36 | 0.92 |
| *PNLIP* | 34 | Pancreas | 33,660 | Ovary | 3.67 | 0 |
| *CPA1* | 31 | Pancreas | 54,500 | Testis | 8.16 | 0 |
| *GP2* | 29 | Pancreas | 14,280 | Prostate | 18.34 | 13.5 |
| *CELA3A* | 24 | Pancreas | 27,130 | Stomach | 14.1 | 0.03 |
| *KRT13* | 20 | Esophagus | 33,960 | Vagina | 13,140 | 3961 |
| *PGC* | 19 | Stomach | 36,720 | Lung | 83.84 | 12.42 |
| *KRT4* | 18 | Esophagus | 22,290 | Vagina | 1375 | 15,069 |
| *PRL* | 17 | Pituitary | 54,500 | Testis | 5.15 | 0 |
| *LIPF* | 15 | Stomach | 29,380 | Testis | 4.33 | 0 |
| *CLPS* | 14 | Pancreas | 51,640 | Stomach | 5.35 | 0 |
| *CTRB2* | 9 | Pancreas | 20,760 | Ovary | 3.21 | 0 |
| *MYBPC1* | 6 | Skeletal muscle | 3587 | Prostate | 54.5 | 21.3 |
| *MYH2* | 6 | Skeletal muscle | 1064 | Colon | 0.52 | 0.02 |
| *ZG16B* | 6 | Salivary gland | 17,540 | Prostate | 48.1 | 187.1 |
| *FGA* | 5 | Liver | 5717 | Stomach | 12.43 | 6.02 |
| *HP* | 5 | Liver | 12,710 | Adipose | 140.1 | 2.7 |
| *CKM* | 5 | Skeletal muscle | 11,138 | Heart | 2987 | 2339 |

Independent study samples were taken from RNA-seq experiments of the 2nd highest GTEx tissue and were not co-sequenced with the highest expressed GTEx tissue. *Data from second highest expressed GTEx tissue.

**Table 2 Allelic inconsistencies found in contaminated samples.**

| Individual | Gene | SNP | Major/minor* | Reads* | Major allele %* | Tissue type** | Reads** | Major allele %** |
|---|---|---|---|---|---|---|---|---|
| GTEX-1 | *KRT13* | rs903 | C/A | 101,908 | 0% | Fibroblast cells | 252 | 50% |
| GTEX-1 | *KRT4* | rs7959052 | T/C | 74,468 | 100% | Fibroblast cells | 203 | 12% |
| GTEX-1 | *KRT4* | rs7956809 | C/G | 85,803 | 100% | Fibroblast cells | 204 | 13% |
| GTEX-1 | *KRT4* | rs2035879 | T/C | 72,978 | 51% | Fibroblast cells | 164 | 7% |
| GTEX-1 | *KRT4* | rs17119475 | G/A | 71,592 | 49% | Fibroblast cells | 226 | 98% |
| GTEX-8 | *CELA3A* | rs9187 | C/T | 162,318 | 73% | Tibial nerve | 1155 | 100% |
| GTEX-8 | *CELA3A* | rs12908 | G/A | 169,394 | 74% | Tibial nerve | 1215 | 100% |
| GTEX-9 | *CELA3A* | rs3820285 | C/G | 98,896 | 1% | Adipose | 5178 | 48% |
| GTEX-9 | *CELA3A* | rs9187 | C/T | 105,462 | 75% | Adipose | 6082 | 97% |
| GTEX-9 | *CELA3A* | rs12908 | G/A | 108,681 | 75% | Adipose | 6313 | 98% |
| GTEX-10 | *CLPS* | rs3748050 | T/C | 80,019 | 47% | Artery | 1117 | 99% |

*Enriched tissue.
**Contaminated tissue.

ubiquitous expression across all tissues. Of 1010 variants obtained from the combined VCF files, 11 had some degree of allelic heterogeneity (Table 2). No incongruencies were found in the 287 variants of the two control genes.

One SNP site, rs7956809, was particularly informative. SNP rs7956809 (C/G), located in *KRT4*, had a relatively low allelic variation, with only five individuals in the entire GTEx cohort homozygous for the alternative allele (G). One individual (arbitrarily GTEX-1) was homozygous C at rs7956809 in both its DNA (VCF file) and matched esophagus (RNA-Seq FASTQ data) (Fig. 1e). However, the rs7956809 SNP in the GTEX-1 fibroblast sample was 87% G and 13% C. Six esophagus samples were sequenced on the same day as the GTEX-1 fibroblast sample. No other esophagus samples were sequenced within 4 days. One of those six samples, GTEX-2, was homozygous G at rs7956809. The five other samples were homozygous C. This strongly implicates the GTEX-2 esophagus sample as the dominant contaminant of the GTEX-1 fibroblast sample.

We further investigated the relationship between the GTEX-1 fibroblast sample and the GTEX-2 esophagus sample finding no clear connection. The two samples were sequenced on different machines and in different flow cells. Of some interest, the sequencing sample adapters (molecular indexes) were similar (Supplementary Data 5).

**The extent of gene contamination in GTEx.** After establishing that contamination exists in GTEx by identification of a temporal association and validation through polymorphisms, we then attempted to address the extent of contamination in the GTEx data set. To characterize this, we investigated the various levels of pancreas enhanced gene expression in non-pancreatic tissue (Fig. 2). In the 11,366 non-pancreas samples investigated, ~25% had 0 reads for each of four pancreas-enriched genes and another ~50% of samples had TPM < 10.

**PEER factor normalization is not fully corrective.** The GTEx analysis pipeline uses probabilistic estimation of expression residuals (PEER) factor to correct for possible confounders[21,22]. This method identifies hidden factors that explain much of the expression variability and can be used to normalize RNA expression data. We focused on just one tissue, lung, and followed

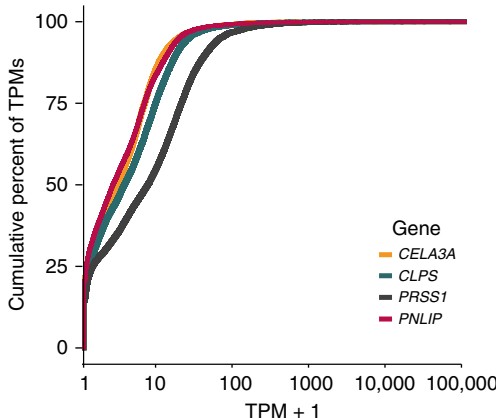

**Fig. 2 A cumulative distribution plot of 11,366 non-pancreas RNA-seq samples and their cumulative TPM expression of four pancreas genes.** The right shift in *PRSS1* is consistent with it having the highest pancreas TPM expression.

the GTEx analysis pipeline to determine the extent to which PEER factor normalization can identify and correct for this contamination. Sixty PEER factors were identified, with the top two capturing a difference between "in hospital" (short post-mortem interval) and "outside of hospital" (longer postmortem interval) deaths (Fig. 3a). This relationship is consistent with our prior report of variation in lung[16]. Similar to the global findings of Fig. 1, *PNLIP* expression was increased in lung samples sequenced on the same day as a pancreas. Despite correcting for 35 or even 60 PEER factors, this difference was not fully accounted for (Fig. 3b). Indeed, of five genes evaluated, only one gene (*KRT4*) was fully corrected for by PEER factors (Table 3). We then explored if this lack of full correction impacted eQTL analysis in the GTEx program.

**Contamination affects GTEx eQTL reporting.** Using the GTEx eQTL browser, we identified 72 tissues reported as having significant eQTLs for the 18 genes listed in Table 1. Seven tissues matched the known dominant expression patterns of the genes. An additional 34 tissues were deemed possible based on expression patterns noted by RNA and protein immunohistochemistry in which expression (in TPM) was above the basal level of all tissues. However, 31 inappropriate tissues were identified as harboring eQTLs even though these genes are not natively expressed in these tissues, appearing only as a result of contamination (Table 4).

**Non-GTEx bulk sequencing data sets confirm contamination.** To determine whether highly expressed tissue-enriched contamination is a feature of sequencing in general, we identified two additional RNA-Seq data sets containing multiple organs/tissues[19,23,24]. Neither study sequenced the pituitary, which is the organ with the highest levels of prolactin (*PRL*) expression (Fig. 3c). Both studies performed multiplexed sequencing on Illumina 2000 or 3000 sequencers. These data sets demonstrate the extent of *PRL* expression contamination across six tissues is dependent on the amount of *PRL* expressed in the appropriate tissues (GTEx pituitary median 54,500 TPM and uterus median 4.01 TPM (Fig. 3d). To additionally characterize the breadth of contamination in the literature, we identified 10 studies with two or more organs/cells sequenced together. All 10 had cross-sample contamination of the type described herein (Supplementary Data 6). We also analyzed the METSIM study, containing bulk sequencing of adipose tissue in 434 subjects[25]. The data from that

study demonstrates how, in the absence of sequencing at the same time as a contaminating tissue (ex. stomach, pancreas), contaminating reads are minimal and do not correlate. This was the opposite of what was observed in GTEx adipose tissue (Supplementary Fig. 3).

**scRNA-Seq is similarly impacted by contamination.** We then investigated this type of highly expressed gene contamination in scRNA-Seq. Although doublets are well-characterized, and barcode swapping has been reported, other forms of contamination are not well-described[12,26]. Also, owing to the low depth of sequencing in current scRNA-seq compared with bulk sequencing, the type of contamination concerned herein would not be expected to generate large numbers of contaminating reads. However, investigations across two cell types, endothelial cells, and mesenchymal cells (fibroblasts, activated stellate cells, cancer-associated fibroblasts), suggested otherwise.

Three different scRNA-Seq data sets (GSE84133, GSE103322, GSE72056) from normal pancreas, metastatic melanoma, and head and neck squamous cell carcinoma were used[27–29]. From the pancreas sample, insulin (*INS*) represented a highly expressed, cell type-enriched gene that is exclusive to beta cells (TPM value of 167,144; transcripts per 10,000 reads (TP10K) = 1671). *INS* expression was modestly elevated in endothelial cells and mesenchymal cells from the same data set (GSE84133), but is absent in endothelial and mesenchymal cells from the other data sets in which beta cells were not sequenced (Fig. 3e). Contamination of pancreatic endothelial cells is further supported by a recent snATAC-seq study that demonstrated closed chromatin at the *INS* locus in pancreatic endothelial cells[30].

**scRNA-seq contamination drives aberrant cell clustering.** Tabula Muris is a single cell transcriptome atlas of 20 mouse tissues[31]. We gathered endothelial cell data from nine tissues and generated tSNE plots of this data based on the inclusion of either 6 or 10 principal components (PCs) (Supplementary Fig. 4a, b). With 10 PCs, a subset of pancreatic endothelial cells incorrectly clustered separately as a result of contaminating pancreatic acinar genes that drive PCs 8 and 10. HPA staining validated the localization of these proteins to acinar cells (Supplementary Fig. 4c).

## Discussion

The GTEx data set represents an ideal resource to study sequence contamination. Its 11,000+ samples from 700+ individuals from a diverse set of tissues with all library preparation and sequencing performed at one center is unique. During our initial variation analysis of 48 tissues spanning 10,294 samples, we detected a variable signal of pancreas genes in 26 of those tissues. From there we noticed genes that were highly expressed in esophagus, stomach, pituitary, and other tissues also appeared in shared clusters across unrelated tissues. These highly expressed, tissue-enriched genes were found at low, variable levels in other organs and represented some of the most frequent causes of variation between samples of the same tissue type. Although many of the genes derive from the pancreas, that is more a feature of their being so highly expressed rather than a feature of the organ.

We found that contamination is best linked to the date of sequencing for both pancreas genes and esophagus genes (linear regression, $p = 1.3e-14$ and $p = 6.1e-49$, respectively). However, both owing to contamination being noted in some samples that are sequenced a few days apart from a possible contaminating source and the SNP-based evidence, we suspect the majority of the contamination occurred during library preparation rather than the sequencing itself. Library preparation dates were not publicly documented.

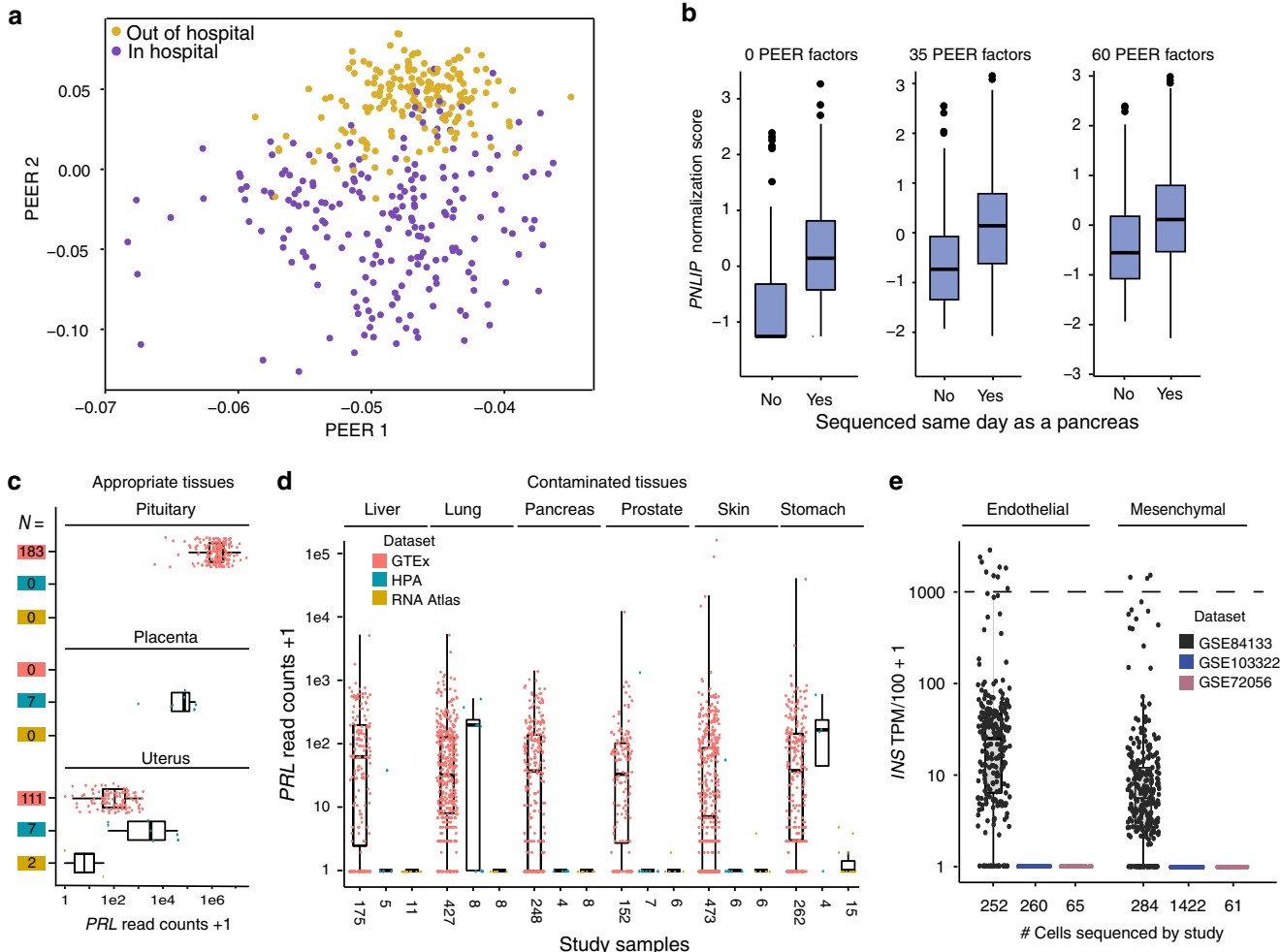

**Fig. 3 Impact of PEER factors on contamination and differing contamination outcomes by study. a** The top two PEER factors separated in hospital from out of hospital deaths ($N = 427$). **b** With no PEER factor correction there is a significant increase in *PNLIP* expression normalized scores in lung samples if they were sequenced on the same day as a pancreas (no=96, yes=331; linear regression, $p = 4.34e-14$). After 35 ($p = 1.38e-11$) or 60 ($p = 3.03e-06$) PEER factor corrections, the difference remained. The solid line in all boxplots represents the median of the data, whereas the lower and upper hinges correspond to the 25th percentile and 75th percentile, respectively. The whiskers represent the interquartile range × 1.5, and any outliers beyond the whiskers are represented as dots. **c** *Prolactin (PRL)* read counts in pituitary (high), placenta (medium), and uterus (low), where *PRL* is known to be expressed across GTEx, HPA, and the RNA Atlas. The numbers in colored boxes indicate sample sizes and the color indicates respective study. **d** *PRL* contamination reads across six tissues from three studies that correlate with levels of likely contamination based on the other sequenced organs. **e** *INS* contamination across three scRNA-Seq data sets. Only in the pancreas data set (GSE84133), where beta cells were also sequenced, does *INS* appear to be lowly expressed in endothelial and mesenchymal cells. Cells with expression above the dotted line at 1000 TP10K are likely doublets or multiplets.

**Table 3 Significance of same-day sequencing of lung with contaminating tissues on gene expression.**

| Gene | P value before PEER correction | P value after correcting for 35 PEER factors | P value after correcting for 60 PEER factors | Beta estimate after correction |
|------|-------------------------------|----------------------------------------------|----------------------------------------------|-------------------------------|
| *PNLIP* | 4.34e-14 | 1.39e-07 | 1.94e-05 | 0.49 |
| *PRSS1* | 6.29e-14 | 5.83e-07 | 2.63e-05 | 0.48 |
| *CELA3A* | 5.91e-14 | 3.36e-07 | 2.41e-05 | 0.49 |
| *KRT4* | 0.0034 | 0.17 | 0.15 | 0.18 |
| *KRT13* | 8.18e-17 | 0.00048 | 0.0036 | 0.37 |

*P values are shown before and after PEER correction.*

Although the nucleic acid isolation date was only modestly associated with contamination, physical contamination can easily occur at this stage. GTEx RNA isolation was manually done in batches of 12 tissues, purposefully with a mix of donors and tissues to minimize batch effects. Samples were individually cut and placed into cryovials for homogenization, followed by further manipulations[32].

At the stage of library preparation or sequencing where our data indicate most of the contamination occurred, there are multiple steps that could be implicated. The library preparation

| Genes | Appropriate tissues | Possible tissues | Inappropriate tissues |
|---|---|---|---|
| *PRSS1* | — | Small intestine | Liver, coronary, skin, lung |
| *PNLIP* | — | — | — |
| *CPA1* | — | — | Coronary |
| *GP2* | — | — | Brain |
| *CELA3A* | Pancreas | Stomach | Liver |
| *KRT13* | Vagina | Lung | — |
| *PGC* | — | Lung, pancreas | Tibial artery |
| *KRT4* | Esophagus | Skin, lung | Colon, brain, thyroid |
| *PRL* | — | — | Gastroesophageal junction, skin, tibial artery |
| *LIPF* | Stomach | — | — |
| *CLPS* | Pancreas | — | — |
| *CTRB2* | Pancreas | — | Aorta, brain, lung, thyroid |
| *FGA* | Liver | Stomach | — |
| *HP* | — | Whole blood, adipose (2), artery (3), lung, tibial nerve, heart | Brain, esophagus mucosa |
| *CKM* | — | — | Aorta, whole blood |
| *MYBPC1* | — | Heart, testis, colon, prostate, brain (2) | Esophagus (2), lung, thyroid |
| *MYH2* | — | — | Colon, lung |
| *ZG16B* | — | Skin (2), adrenal, fibroblasts, lung, stomach, prostate, spleen, colon, testis, whole blood | Adipose, esophagus, pituitary |

**Table 4 Distribution of GTEx eQTLs by tissue type in contaminating genes.**

was completed automatically in 96-well plates with a mix of tissues and individuals to prevent batch effects[32]. Fluidic carryover could have occurred here. At the sequencing level, a major concern is index contamination where index oligonucleotides used for multiplexing can ligate to other sample transcripts, thus contaminating the data after demultiplexing. Index based contamination is machine and lane specific and can even occur at the creation of the indexes when multiple indexes are purified on the same high-performance liquid chromatography column[33]. In addition, if steps to clean libraries of free adapters/primers are not properly executed, the remaining indexes can contaminate clusters in the flow cells. Molecular recombination of indexes during sequencing can also lead to faulty read assignment as multiplex clusters can become contaminated by other samples that acquire the indices of the native sample (index hopping). This has been demonstrated in scRNA-Seq, and seems to occur more often on a HiSeq 4000 platform[26].

Specific to this study, GTEx samples were run on either HiSeq 2000 or 2500 machines in high output mode using cbot clustering. The flowcells were not patterned, and therefore not prone to index-swapping (personal communication, GTEx Help Desk). GTEx's use of custom i7 (later Illumina kit-based) dual indices also reduced the amount of index hopping that can occur[32,33].

Contamination of scRNA-Seq, may or may not be occurring by yet an additional mechanism. Ambient RNA, from disrupted ("broken") cells may be a major source of contamination, as recently described[34]. If true, this would occur before the library preparation and sequencing steps, would cause only a modest expression of the gene, and would not be detectable by current doublet correcting mechanisms[12,35]. This method of contamination might have implications for Human Cell Atlas studies, as we demonstrated it clearly affects cross-tissue comparisons of common cell types in Tabula Muris[31,36].

Using other bulk RNA-Seq data sets with similar sequencing methods and scRNA-Seq data sets, with different preparation methods, we validated that it is predominately contamination, not low-level transcription, which results in non-zero expression values in inappropriate tissues/cells (Fig. 3d, e, Supplementary Fig. 3, Supplementary Fig. 4, Supplementary Data 6). These extra data sets also uncover the generalizability of this highly expressed transcript contamination regardless of the labs in which they take place or the methods employed.

So how big is the contamination problem? It depends on how one uses the data. Fortunately, in the GTEx data, the levels are overall low with only 2.85% of samples having relatively high levels of *PRSS1* (TPM > 100). Thus, for many uses of GTEx data, this level is irrelevant. However, for experiments that involve differential expression profiling, these genes will repeatedly appear due to their variable levels of contamination. We additionally note that the GTEx standard normalization pipeline using PEER factors did not entirely eliminate this source of variation and numerous eQTLs that were identified for the 18 genes described herein were located in incorrect tissues (>40%). We caution that our results do not suggest that one should design a study in which sample type (e.g., tissue type or disease state) is perfectly confounded with library preparation or sequencing date. In fact, it was only because the GTEx study sequenced samples from different individuals on the same date that we were able to definitively show that contamination between samples occurred.

In other scenarios, this basal contamination may be more worrisome. Many publications have reported rare, but variable gene expression in their samples claiming their importance or disease-related behaviors[37]. Our findings call these reports into question. The extent of cross-contamination, where one laboratories' samples are prepped and sequenced at the same time as a different laboratories' unrelated samples through a university core sequencing facility or sequencing company is unknown, but likely frequent[38,39]. The xenomiR story, that rice miRNAs are found in human blood through dietary means[40], was shown to result from library preparation contamination[41,42]. For scRNA-Seq, this contamination falsely implies some gene expression is ubiquitous across cells, which influences computational heterogeneity analyses of cell types, beyond the known challenges of other biological and technical artifacts[31,34,43]. Also, our work highlights the fact that work flows must be considered carefully in very-low DNA mutation detection analysis in clinical cancer samples as samples with higher tumor burdens may contaminate samples with lower tumor burdens and falsely suggest treatment approaches[44,45]. In particular, GTEx data are available in many outlets, including the UCSC Genome Browser. Variable, low-level expression of *PRSS1*, *CELA3A*, and others may falsely intrigue researchers, particularly within the reported eQTLs.

We described low-level, variable expression contamination in the GTEx RNA-Seq data set. This variation was most noticeable

for 18 highly expressed, tissue-enriched genes and strongly correlates with the library preparation and sequencing of the samples. Similar contamination was observed in other bulk and scRNA-Seq data sets, suggesting a universality to this type of contamination. Evaluating low-level variable gene expression in RNA sequencing data sets must be performed with precaution and awareness of potential sample contamination.

## Methods

**Ethics statement**. All human data were publicly available or used with approval of the GTEx consortium. Consent was obtained by those studies.

**Retrieval of GTEx data sets, FASTQ files, and sample data**. The gene read counts of the RNA-Seq GTEx version 7 data set (GTEx_Analysis_2016-01-15_v7_RNASeQCv1.1.8_gene_reads.gct.gz) were downloaded from the GTEx Portal (https://gtexportal.org/home/datasets), along with the de-identified sample annotations (GTEx_v7_Annotations_SampleAttributesDS.txt). The FASTQ files of the tissue samples and the VCF files of appropriate individuals were downloaded from dbGaP (phs000424.v7.p2) with the required permissions.

**Retrieval of human protein atlas tissue-enriched gene list**. We obtained the HPA tissue-enriched genes by downloading a CSV file from this filtered site (https://www.proteinatlas.org/search/tissue_specificity_rna:any;Tissue%20enriched+AND+sort_by:tissue+specific+score, visited on 6/21/18).

**Second highest tissue expression of tissue-enriched genes**. Using the GTEx portal (v7) we noted the TPM for the second highest expressing tissue with >60 samples. We then used GEO, SRA Run Selector, and Recount2 (https://jhubiostatistics.shinyapps.io/recount/) to find data sets where only one tissue was sequenced, with the exception of the ovary sample, which was obtained from the Illumina BodyMap (16 tissue types, absent of pancreas)[46]. Owing to our inability to find salivary gland data we used prostate tissue, the third highest expressing tissue for *ZG16B*. Heart (ERP009437), adipose tissue (SRP053101), lung (SRP032833), prostate (SRP003611), stomach (ERP010889), and colon (SRP029880) TPMs were acquired through Recount2 and were selected for normal samples. The testis data were downloaded from GEO (GSE103905). Vagina and ovary FASTQ data were downloaded from SRA run selector (GSE68229 and GSE30611). FASTQ files were mapped to the Genome Reference Consortium Human Build 38 (GRCh38) using the software HISAT2 version 2.1.0[47]. The output SAM files were turned into BAM files and indexed using samtools version 1.9[48,49]. Assembly was completed using StringTie v1.3.4d and StringTie TPMs were used from the output[50,51].

**Bulk sequencing processing**. The acquired raw read counts were segmented into separate tissue subsets (48 tissues with ≥70 samples each) and their read counts were normalized using the VST feature in DESeq2 version 1.22.1 in R version 3.6.1[17]. This method incorporates estimated size factors based on the median-ratio method, and transformed by the dispersion-mean relationship. We then filtered the 56,202 genes based on their mean expression (mean transformed count >5) to reduce noise and lessen the inflated effect of low expressing genes on correlations.

**Identification of highly variable genes and clusters**. All analyses were completed in R version 3.6.1. In each tissue, a threshold of a >4 variance of VST normalized read counts was used as our cutoff for highly variable transcripts. These genes were then clustered using hierarchal clustering on a distance generated by 1−Kendall's rank-correlation coefficient. A tau critical value was calculated based on the number of samples and genes expressed. The correlation-based dendrogram was cut to produce gene clusters with average within cluster correlation of at least the tau critical value.

**Calculation of normalized expression scores**. Normalized expression scores allow one to summarize the expression of a gene cluster in a sample by the average normalized score of the genes in that cluster. These normalized scores were calculated by subtracting the mean expression and dividing by the median absolute deviation of the expression values for each gene across all samples within a given tissue. The equation is as follows, where $x$ is the VST normalized expression of gene $j$ in sample $i$, $t(i)$ is the tissue type for sample $i$, and $J$ is the number of genes in a given cluster:

Equation 1

$$S_i = \frac{1}{J} \sum_{j=1}^{J} \frac{x_{ij} - \bar{x}_j}{\underset{m:t(m)=t(i)}{\text{Median}} |x_{mj} - \bar{x}_j|} \tag{1}$$

A "pancreas" contamination score was defined as the average normalized score using the genes *PRSS1, CELA3A, PNLIP*, and *CLPS*, across all tissues. An "esophagus" normalized score was defined as the average normalized score using genes *KRT4* and *KRT13* in a given tissue. All normalized scores were calculated

independently for each tissue, to account for tissue-specific between-sample variation in the expression of these genes.

**Linear mixed model analysis for pancreas gene contamination**. Linear models were used to find associations between nucleic acid isolation or sequencing on the same day as a potential contaminating tissue (source of gene expression). These analyses were completed in R version 3.6.1 using the lmer function in package lme4 (v1.1-21). In the linear mixed models we used all available tissues with > 40 samples not sequenced on the same day as a pancreas sample (15 tissues, N = 6258). For the model, tissue was used as a covariate along with "pancreas sequencing day" and "pancreas nucleic acid isolation day" were coded binarily as True/False based on whether a given non-pancreas sample was sequenced or underwent RNA isolation on the same day as a pancreas sample. Our model identified an association between the pancreas contamination score with pancreas sequencing day (T/F; 1/0), pancreas nucleic acid isolation day (T/F; 1/0), and tissue as covariates, with the subject IDs included as a random effect. The equation for our model is below where $t(i)$ gives the tissue type of sample $i$, $s(i)$ is defined as the sequencing day of the sample, and $d(i)$ is the isolation date of sample $i$. At last, $T$ is the number of tissue types, $u_{n(i)}$ is the random effect of subject ID for sample $i$, and $\varepsilon_i$ is the error term for said sample:

Equation 2

$$Y_i = \beta_0 + \beta_1 I\left(s(i) \in \left\{s\left(m : t(m) = t_{\text{pancreas}}\right)\right\}\right)$$
$$\beta_2 I\left(d(i) \in \left\{d\left(m : t(m) = t_{\text{pancreas}}\right)\right\}\right)$$
$$\sum_{k=3}^{T+1} \beta_k I(t(i) = k - 1) + u_{n(i)} + \varepsilon_i \tag{2}$$

**Linear mixed model analysis for esophagus gene contamination**. We repeated the above analysis with the esophagus mucosa, coding shared sequencing day and shared nucleic acid isolation day similarly. In the linear mixed model we used all tissues with a sample size > 40 samples not sequenced on the same day as the esophagus mucosa (8 tissues, N = 3917). The model used esophagus nucleic acid isolation day (T/F; 1/0), esophagus sequencing day (T/F; 1/0), and tissue as covariates along with subject IDs as a random effect on the esophagus contamination score.

**Base pair incongruency analysis**. Base pair incongruency analysis required a contaminated tissue expression FASTQ, a native tissue expression FASTQ, and the individual's VCF file. FASTQ files were mapped to the Genome Reference Consortium Human Build 37 (hg19) using the software HISAT2 version 2.1.0[47]. The output SAM files were turned into BAM files and indexed using samtools version 1.9[48,49]. Preliminary analysis and development of figures were generated using the Integrative Genome Viewer version 2.4.13[52,53]. Protein coding SNPs, rare variants, and personal variants (collectively referred to as variants in this paper), were manually selected using IGV as a reference. Using the tool bam-readcount version 0.8.0 in combination with a Python 3.6.2 script, a list of RNA-Seq and genomic incongruencies were generated for the acquired sample BAM files.

**PEER factor analysis**. We obtained the GTEx RNA-Seq data set from lung (N = 427). The data underwent trimmed mean of m-values (TMM) normalization and filtering out of lowly expressed genes (< 0.1 TPM for 80% or more of the samples) before running PEER to identify potential confounders[21]. Following GTEx's pipeline (https://gtexportal.org/home/documentationPage#staticTextAnalysisMethods visited 02/24/2020), we then performed an inverse-normal transformation on the expression values for each gene in order to reduce the effect of outliers[22]. Normalized scores for each gene are based on TMM-normalization, inverse-normal transformation, and scaling/centering at zero using the base R scale function. A linear regression was performed, with either raw inverse-normalized expression values or expression normalized scores (after correcting for PEER factors) as the outcomes and same-day pancreas (/esophagus) sequencing status as the predictor. Beta estimates represent how many standard deviations greater the mean expression of a gene is when samples are sequenced on the same day as a contaminating tissue, even when accounting for variance explained by 60 PEER factors.

**Cross-referencing eQTLs with contamination findings**. We obtained and tallied eQTL reports for the 18 genes in Table 1 from the GTEx eQTL browser (https://gtexportal.org visited on 26 March 2019). eQTLs were identified by tissue association and conservatively placed in one of three categories: appropriate expression, possible expression, and inappropriate expression. The appropriateness of expression in any tissue was based on the evaluation of TPM levels in the tissue and immunohistochemistry staining patterns as noted in the HPA[54].

**Acquiring HPA, RNA Atlas RNA-Seq, and METSIM Data**. Using the R package recount version 1.8.2, we downloaded HPA RNA-Seq data, accession ERP003613[19]. The RNA Atlas was acquired by downloading their raw RNA-seq counts from Gene Expression Omnibus (GSE120795)[23]. The HPA RNA-Seq was performed across 27 tissues and the RNA Atlas was across 20 tissues. We filtered

samples down to the shared tissues of liver, lung, pancreas, prostate, skin, and stomach as well as the potential sources of *PRL* contamination of pituitary, placenta, and uterus. Only sun exposed skin was used for GTEx analysis. TPM RNA-seq data from the METSIM study was downloaded through the GEO (GSE135134)[25].

**Acquiring and normalizing scRNA-Seq data.** We identified three publications with human scRNA-Seq data sets that all contained endothelial and mesenchymal cells[27–29]. All three sets had median read counts/cell >100,000. The processed read counts from the three studies were obtained from the Gene Expression Omnibus (GEO) (GSE84133; GSE103322; GSE72056). For each study we used the supplied cell type information to identify endothelial and mesenchymal-type cells. Mesenchymal cells were labeled as "activated stellate" (GSE84133), "fibroblast" (GSE103322), and "cancer-associated fibroblast" (GSE72056). The GSE103322 and GSE72056 read data were generated using Smart-Seq2 on a NextSeq 500 instrument[55]. Raw FASTQ data sets were normalized using RSEM to calculate $E_{i,j} = log_2(TPM_{i,j}/10 + 1)$, where $TPM_{i,j}$ refers to transcript-per-million for gene $i$ in sample $j$[56]. This was done to create TP10K values to account for single cell read depth and the addition of 1 was to limit the effect of 0 s in downstream analysis[27]. GSE84133 was generated using the inDrop method and sequenced on a HiSeq 2500 instrument[57]. Human islets were isolated using a modified Ricordi method, which includes collagenase, mechanical agitation, pumping, and centrifugation[58]. To make single cells, the islets were then centrifuged twice at 250 rpm and treated with TrypLE Express before mechanical dispersion with a P100 pipette, followed by centrifugation at 500 rpm. The two cancer sample libraries (GSE103322, GSE72056) were generated using Smart-Seq2 and sequenced on a NextSeq 500 instrument[55]. Read counts were scaled using TP10K to equalize expression level counts to the other two studies. The calculation of the percent contamination of endothelial cells by *INS* was the ratio of average normalized *INS* read counts per endothelial cell divided by the normalized average read count in beta cells and excluding the nine cells with >1000 TP10K.

We obtained the Tabula Muris read data from https://figshare.com/articles/Robject_files_for_tissues_processed_by_Seurat/5821263[31] (visited on 8/8/19)[31]. We used the R package Seurat (v3.1.1) and selected endothelial cells from nine organs/tissues (brain, fat, heart, kidney, limb muscle, liver, lung, pancreas, trachea) based on the annotation of the seurat object in the metadata slot cell_ontology_class labeled "endothelial cell"[59]. tSNE plots were generated based on the top 6 or top 10 PCs. The average number of PCs used by Tabula Muris per tissue/organ to create tSNE maps was 12 and the median was 10. Images of protein expression of contaminating pancreas genes were obtained from the HPA[54].

**Reporting summary**. Further information on research design is available in the Nature Research Reporting Summary linked to this article.

## Data availability
No new sequencing data were created for this study. Sequencing data used in this study is available through dbGap (https://www.ncbi.nlm.nih.gov/gap/): GTEx (performed with phs000424.v7.p2; now phs000424.v8.p2); GEO (https://www.ncbi.nlm.nih.gov/gds): Vagina FASTQ (GSE68229), Ovary FASTQ (GSE30611), Testis RNA-seq (GSE103905), RNAAtlas (GSE120795), METSIM Study (GSE135134), Endothelial/Mesenchymal Single Cell Studies (GSE84133, GSE103322, GSE72056); recount2 (https://lcolladotor.github.io/project/recount2/): Heart RNA-seq (ERP009437), Adipose RNA-seq (SRP053101), Lung RNA-seq (SRP032833), Prostate RNA-seq (SRP003611), Stomach RNA-seq (ERP010889), Colon RNA-seq (SRP029880), HPA RNA-seq (ERP003613); or Tabula Muris (https://figshare.com/articles/Robject_files_for_tissues_processed_by_Seurat/5821263) (GSE109774). All data are available from the corresponding author upon reasonable request.

## Code availability
Code for all analyses is deposited at GitHub (https://github.com/mhalushka/gtex_contamination_code). Computational analyses were done using public R packages except when specifically noted otherwise.

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

## Acknowledgements

We thank the GTEx Help Desk, Kristen Ardlie, Sheila Dodge, and NDRI for valuable assistance in understanding the technical data sets and islet isolation methodology. M.K.H. was supported by grants 1R01HL137811, R01GM130564, and P30CA006973 from the National Institutes of Health and 17GRNT33670405 from the American Heart Association. T.O.N. was supported by grants R01GM130564 and T32GM07814. D.E.A was supported by 1R01HL131573 and 1R01HL137811. M.N.M. was supported by R01HL137811 and the University of Rochester CTSA award number UL1TR002001. A.Z.R. was supported by R01GM130564.

## Author contributions

M.K.H., M.N.M., and A.Z.R. conceived of the experiments and assisted with the manuscripts. T.O.N. performed the experiments, analyzed the data, and wrote the manuscript. S.Y., R.X.V., V.P., and D.E.A. performed experiments and assisted on the manuscript.

## Competing interests

The authors declare no competing interests.
