## [Peer Review File · Nature Communications]

Reviewer #1 (Remarks to the Author):

Nieuwenhuis and colleagues illustrate that cross-sample contamination occurs in the GTEx data as well as other multi-tissue data sets. The main focus is on a set of highly expressed genes enriched in pancreas or esophagus, which can be found in other tissues, at levels correlating with whether the sample was sequenced on the same day as a sample from the contaminating tissue. Overall, I think the authors provide convincing arguments that such contamination indeed occurs in GTEx, and go into some detail to understand in which step of the process it happens. Below I provide more specific comments.

1. Most of the genes identified as "highly expressed, tissue-enriched, and highly variable in at least 5 other GTEx tissues", and especially those analyzed further, are pancreas genes. Could the authors provide more information as to why genes specific to other tissues are less abundant in this category - are there less tissue-enriched genes to begin with, are they not expressed highly enough, or are they not highly variable in other tissues (and if the latter, could they still be highly expressed in these other tissues)? My main concern here is that, at least from the information given, there doesn't seem to be an obvious reason to believe that it's more likely for a pancreas sample to contaminate other tissues, than it is for samples from other tissues.

2. It would be interesting to see to what extent the variance is indeed approximately independent of the mean after the variance stabilizing transformation has been applied. If not, the selection of highly variable genes would also be influenced by the expression level of the genes.

3. It was not clear to me how the linear regressions with nucleic acid isolation date and/or sequencing date were done. Was the actual date used as a predictor, and if so, is it a reasonable assumption that the effect of time on gene expression is linear? On line 147 it is stated that a modest association was found between nucleic acid isolation date and "presence of contamination". How was the presence of contamination encoded (as a categorical variable)? Similarly, it is not clear what it means that "contamination was 0.85 standard deviations higher when a sample was sequenced on the same day as a pancreas sample" (higher compared to what baseline?). Overall, I feel that the language concerning the statistical modeling would benefit from clarification.

4. In Table 5 (on eQTLs), it is indicated that some tissues, in addition to the tissue with the highest expression, "possibly" express the studied genes. Could this be seen also in the previous gene expression analysis; i.e., could the mix of "possible" and "inappropriate" tissues in the overall analysis affect the conclusions there? It is not clear whether the genes would show up as highly variable in the "possible" tissues, or if they would rather be constitutively expressed there.

5. The paper states that analyses were made with R v3.5.1, but the reporting summary states v3.6.1. No version information for R packages is given. data-table is not a valid R package name (should be data.table).

6. Several items used in the paper are listed as n/a in the reporting summary. For example, test statistics

(for the linear regression) with confidence intervals, effect sizes, degrees of freedom and p-values, and a full description of statistical parameters and other basic estimates. These should be provided for the linear regression analyses in the paper.

7. The code used for the analyses should preferably be made available to readers.

8. Why were CPM values used for the "second highest tissue expression", rather than TPM values, which would be more comparable to the GTEx TPM values? I think that also regarding expression units (counts, TPMs, VST expression values), the text could be clearer/more precise. For example, it is not clear to me what is meant by "Read counts were scaled using TPM/100 to equalize expression level counts to the other two studies."

Reviewer #2 (Remarks to the Author):

In this work, the authors analyze RNA-seq data from GTEx tissues and identify signs of potential sample contamination such as observing co-variation clusters of genes highly and specifically enriched in pancreas in 24 unrelated tissues. In a linear model, sample contamination was significantly associated with a sample of the native tissue expressed on the same day. The authors also analyze genotype differences from DNA and RNA. Also presented are analyses considering contamination implications in eQTL and scRNAseq data. While the study is interesting and quite relevant to the field, I have a few questions as I try to interpret the results, answering/clarifying which could make the study more robust:

1. Fig 1A – what if the genes in the ‘contamination’ cluster are just lowly and variably expressed across individuals in different tissues, or are due to technical artifact such as PCR duplicates instead of widespread cross-sample contamination (especially because the authors describe observing the pancreas genes in as many as 24 tissues)? To directly answer this, one could utilize independent eQTL studies that performing RNA-seq on samples from one tissue, say, subcutaneous adipose eQTL study (10.1016/j.ajhg.2019.09.001, N=434) and run a similar analysis as Fig 1A. This should enable confirming that these samples do not show expression or co-variation of the genes in the ‘contamination’ cluster? A comparison could also be useful for the eQTL analysis in the manuscript. While the authors compare other datasets in Fig 2d, I’m concerned about a large difference in the number of samples; large deeply sequenced RNA-seq studies are more likely to identify more lowly expressed genes.
2. Table 1 – Why isn’t TPM reported for independent study for the 2nd highest tissue for a direct comparison?
3. Fig 1a – there appear to be some sub-clusters in the ‘contamination’ cluster – do these correspond to different tissues based on the genes highlighted?
4. Fig 1b,c – show that the Z score of the gene PRSS is higher in tissue samples where a pancreas sample was sequenced the same day. It would be reassuring to see violin+jitter plots of TPM of the co-variation cluster genes (PRSS, PNLIP, CLE3A etc) in the 24 tissues where this cluster is observed vs the other 24 where it is not observed as control, classified into if a pancreas sample was sequenced the same day or not. It would be useful to see plots for other genes highly expressed in the pancreas (eg. INS) not

identified in the co-variation cluster if still there is some trend with the sequencing day. Fig 1c – a Wilcoxon Rank Sum Test would aid in quantifying the trend in this plot.

5. For the allelic incongruency analysis, are the authors accounting for read quality?

6. GTEx 1 fibroblast and GTEx2 esophagus comparison in an interesting example. Were there SNPs in any other genes with detectable expression in fibroblasts that could support contamination? i.e. homozygous genotype in fibroblast but robustly detectable alternate alleles in RNA?

7. Table 3 mentions read counts – it would be more robust to use TPM here to account for different read depths across samples. Such a table could also be better visualized as a cumulative distribution plot across TPM thresholds.

Other minor comments:

Line 91 – is there a citation missing?

Lines 97-108 describe an analysis, it would be helpful to the readers if the appropriate figure was referenced

Figure 1A and Supplementary Fig 1, it would be more useful to the readers if some genes in other co-variable clusters were also labeled.

Line 146 – The authors go from co-variation clusters to regression model including isolation date – it would be helpful if the authors clarified in this sentence what was the metric to quantify contamination for each sample.

Line 146 – were all tissues considered together in the linear regression model?

Line 103-105 is there a citation?

Line 214 – table reference is wrong

Line 293 – figure reference should be 2e?

Reviewer #3 (Remarks to the Author):

The idea behind this paper is highly innovative, and brings to light an important issue. However, the manuscript lacks sufficient detail on a number of aspects of the study materials and methods that need to be addressed.

One major concern is the lack of information about the sequence runs with goal of more precisely localizing the source of the cross-contamination to a given stage of the sequence run. These comments are included in the comments to the bioRxiv publication, and I think need to be addressed: "Currently the text states that they were all Illumina HiSeq 2000/2500 runs. But the HiSeq 2500 could run in one of two "modes" using either standard "High Output" or "Rapid" chemistry. "Rapid" chemistry could use either "on-board" or cBot clustering. On board clustering trafficked the contents of the run library pool to both lanes of the flow cell through fixed (non-disposable) tubing and this was known to be a source of run-to-run contamination as, to my knowledge, no bleach wash protocol was ever offered by Illumina for the 2500. Whereas high output chemistry pumped lane pools of libraries through disposable tubing while clustering them. This occurred on a separate instrument, called a cBot.

For a large project of this sort, one might tacitly presume that the chemistry used would be High Output.

But it is also mentioned in the text that dual indexing was used for this project. Dual indexing was not "native" to the 2000/2500 high output chemistry kits -- requiring the addition of custom primer for i7 indexing. If high output (cBot clustered) flow cells were used, then any lane-to-lane or run-to-run contamination detected would likely derive from processing prior to clustering.

This would ultimately get at the question as to whether this cross-contamination is the result of index hopping -- a phenomenon largely ascribed to newer Illumina instruments that use patterned flowcell and x-amp clustering chemistry. Another possibility to consider would be whether Illumina's default mismatch=1 indexing was used during demultiplexing and if much of the contamination could be avoided by setting it to mismatch=0."

Additionally, detail should be added to the section at Page 5, line 110-113: (also in the 2nd column of Table 1). They demonstrate that these genes were highly expressed in their native tissues, but how were they identified as variables in unrelated tissues should be discussed. At least, the name of used method should be mentioned. I think this key step ought to be as clear as possible so that other people can reproduce the work.

On a more minor note,

Fig 1a: Why are these genes grouped in the contamination cluster? Because these genes are not supposed to show up in these tissues?

Page 8, line 146-149: Since the authors said they used linear regression, why not list what equation they really used. What was the dependent variable in the model? Was the nucleic acid isolation date the only independent variable? Is there any covariate? What was the p-value and estimate for intercept term? There is more information needed here to support the conclusion.

Additionally, it would be better to use model comparison instead of p-values from linear regression for this purpose.

The p-value is $1.21e-67$ in Fig 1b (Page 7, line 123), but $2.66e-75$ in Page 8, line 154.

On a related note, why not conduct t-test between these two groups?

The meaning of y-axis in Fig. 2c is not clear.

Typos: Page 3, line 64: Limitations exists should be exist.

Reviewers' comments:

Reviewer #1 (Remarks to the Author):

Nieuwenhuis and colleagues illustrate that cross-sample contamination occurs in the GTEx data as well as other multi-tissue data sets. The main focus is on a set of highly expressed genes enriched in pancreas or esophagus, which can be found in other tissues, at levels correlating with whether the sample was sequenced on the same day as a sample from the contaminating tissue. Overall, I think the authors provide convincing arguments that such contamination indeed occurs in GTEx, and go into some detail to understand in which step of the process it happens. Below I provide more specific comments.

1. Most of the genes identified as "highly expressed, tissue-enriched, and highly variable in at least 5 other GTEx tissues", and especially those analyzed further, are pancreas genes. Could the authors provide more information as to why genes specific to other tissues are less abundant in this category - are there less tissue-enriched genes to begin with, are they not expressed highly enough, or are they not highly variable in other tissues (and if the latter, could they still be highly expressed in these other tissues)? My main concern here is that, at least from the information given, there doesn't seem to be an obvious reason to believe that it's more likely for a pancreas sample to contaminate other tissues, than it is for samples from other tissues.

The reviewer is correct that our data is weighted towards the pancreas. The overall argument of the manuscript is that the most highly expressed genes (by TPM or similar measurement) are going to be the most obviously detected as contaminating reads have TPMs ~1500 fold lower in affected organs. It is apparently a quirk of nature that pancreatic enzyme genes are so highly expressed. Should a different tissue have a similar number of highly expressed, tissue-specific genes that would be another organ that we would have focused on.

To demonstrate this, we obtained the most highly expressed genes (by TPM) from all of the GTEx tissues and show the top 15 here in tabular form. As you can see, the pancreas has 5 of the top 15 highest expressed genes, thus our focus on that tissue. The pituitary and stomach have two each. Genes listed as ubiquitous have low expression profiles across multiple tissues, despite their being dominant in one tissue (ex *HBB* in whole blood or *ACTA1* in skeletal muscle).

Count	Gene	TPM	In this study	Tissue
1	HBB	246600		Ubiquitous
2	PRSS1	99095	Yes	Pancreas
3	HBA2	82890		Ubiquitous
4	REG1A	57225	Yes	Pancreas
5	CPA1	54505	Yes	Pancreas
6	PRL	54500	Yes	Pituitary
7	CLPS	51645	Yes	Pancreas
8	GH1	48790	Yes	Pituitary
9	PGC	36720	Yes	Stomach
10	S100A9	34070		Ubiquitous

11	ACTA1	34000		Ubiquitous
12	KRT13	33960	Yes	Esophagus
13	PNLIP	33655	Yes	Pancreas
14	POMC	32270		Ubiquitous
15	LIPF	29380	Yes	Stomach

2. It would be interesting to see to what extent the variance is indeed approximately independent of the mean after the variance stabilizing transformation has been applied. If not, the selection of highly variable genes would also be influenced by the expression level of the genes.

To answer this question, for several tissues we have plotted the standard deviation and rank order of mean expression of genes that passed our filter of >5 variance stabilizing transformation (VST) normalized counts. As noted by the curve of the red line, indicating the running median estimator, there does not seem to be an increase with variable genes and mean expression. Therefore we have concluded the

variance is essentially independent of the mean.

3. It was not clear to me how the linear regressions with nucleic acid isolation date and/or sequencing date were done. Was the actual date used as a predictor, and if so, is it a reasonable assumption that the effect of time on gene expression is linear? On line 147 it is stated that a modest association was found between nucleic acid isolation date and "presence of contamination". How was the presence of contamination encoded (as a categorical variable)? Similarly, it is not clear what it means that "contamination was 0.85 standard deviations higher when a sample was sequenced on the same day as a pancreas sample" (higher compared to what baseline?). Overall, I feel that the language concerning the statistical modeling would benefit from clarification.

Thank you for bringing up this point to indicate that we were not clear. Specific to your question, the chronological date was not used as a predictor. The coded variables used included "pancreas sequencing day" and "pancreas nucleic acid isolation day" both of which were coded as "do the other tissues share the same sequencing date with a pancreas sample (True/False)." Contamination was not encoded as a categorical variable, it was encoded as a "normalized score" a continuous variable that was derived from our initial analysis. In our new, more robust linear mixed models, we now have separate pancreas and esophagus normalization scores, based on the genes *PRSSI*, *CELA3A*, *PNLIP*, and *CLPS* for pancreas and *KRT4* and *KRT13* for esophagus mucosa. The results of these new linear models are provided as Supplementary Data 3 and 4. Two additional Supplementary Data (1 and 2) show the number of samples sequenced on the same day as a pancreas or esophagus. We only used tissues with >40 samples, to ensure the robustness of our model. New, clearer text is on pages 8-9. Supplementary Data 3 and 4 are shown below.

Linear mixed models:

Supplementary Data 3: The output of the pancreas gene contamination linear mixed model.

	Estimate	Std. Error	Lower CI	Upper CI	df	t value	p value
(Intercept)	-0.757	0.044	-0.844	-0.67	5639.634	-17.15	2.60E-64
Tissue sequenced same day as pancreas (Yes/No)	0.863	0.025	0.814	0.912	6239.197	34.331	9.50E-237
Tissue isolated same day as pancreas (Yes/No)	0.175	0.025	0.126	0.225	5896.169	6.955	3.90E-12
Tissue Type: Adipose - Visceral (Omentum)	-0.073	0.05	-0.171	0.025	5725.538	-1.456	0.145
Tissue Type: Artery - Aorta	-0.126	0.053	-0.229	-0.022	5752.369	-2.387	0.017
Tissue Type: Artery - Tibial	-0.021	0.047	-0.113	0.071	5700.359	-0.45	0.652
Tissue Type: Breast - Mammary Tissue	-0.117	0.053	-0.222	-0.013	5754.809	-2.211	0.027
Tissue Type: Cells - Transformed fibroblasts	-0.023	0.053	-0.126	0.081	5836.869	-0.428	0.668
Tissue Type: Esophagus - Muscularis	-0.173	0.05	-0.27	-0.076	5725.209	-3.488	0.0004
Tissue Type: Heart - Atrial	-0.076	0.053	-0.18	0.027	5754.206	-1.445	0.148

Appendage							
Tissue Type: Heart - Left Ventricle	-0.036	0.052	-0.138	0.067	5738.204	-0.683	0.494
Tissue Type: Lung	-0.035	0.047	-0.128	0.058	5712.541	-0.733	0.463
Tissue Type: Muscle - Skeletal	-0.025	0.044	-0.111	0.062	5668.275	-0.558	0.576
Tissue Type: Nerve - Tibial	-0.035	0.048	-0.129	0.059	5693.583	-0.735	0.462
Tissue Type: Skin - Not Sun Exposed (Suprapubic)	-0.101	0.049	-0.197	-0.005	5721.591	-2.07	0.038
Tissue Type: Skin - Sun Exposed (Lower leg)	-0.029	0.046	-0.119	0.061	5672.47	-0.627	0.53
Tissue Type: Thyroid	-0.081	0.047	-0.173	0.011	5711.562	-1.732	0.083
Tissue Type: Whole Blood	-0.003	0.05	-0.1	0.094	5792.64	-0.064	0.949

Supplementary Data 4: The output of the esophagus mucosa gene contamination linear mixed model.

	Estimate	Std. Error	Lower CI	Upper CI	df	t value	p value
(Intercept)	-1.24	0.059	-1.356	-1.125	3037.809	21.038	7.70E-92
Tissue sequenced same day as esophagus (Yes/No)	1.555	0.039	1.479	1.632	2159.702	39.774	5.00E-260
Tissue isolated same day as esophagus (Yes/No)	-0.027	0.039	-0.104	0.05	3501.761	-0.69	0.49
Tissue Type: Artery - Tibial	0.086	0.051	-0.014	0.186	3415.339	1.695	0.09
Tissue Type: Heart - Left Ventricle	0.064	0.057	-0.047	0.176	3470.893	1.131	0.26
Tissue Type: Lung	0.079	0.051	-0.022	0.18	3429.574	1.535	0.12
Tissue Type: Muscle - Skeletal	0.01	0.048	-0.084	0.103	3370.268	0.2	0.84
Tissue Type: Nerve - Tibial	0.053	0.052	-0.048	0.155	3407.312	1.029	0.3
Tissue Type: Skin - Sun Exposed (Lower leg)	-0.005	0.05	-0.103	0.093	3377.885	-0.097	0.92
Tissue Type: Thyroid	0.003	0.051	-0.097	0.103	3426.607	0.056	0.96
Tissue Type: Whole Blood	0.041	0.056	-0.069	0.152	3575.867	0.733	0.46

4. In Table 5 (on eQTLs), it is indicated that some tissues, in addition to the tissue with the highest expression, "possibly" express the studied genes. Could this be seen also in the previous gene expression analysis; i.e., could the mix of "possible" and "inappropriate" tissues in the overall analysis affect the conclusions there? It is not clear whether the genes would show up as highly variable in the "possible" tissues, or if they would rather be constitutively expressed there.

This is a very thoughtful question and reveals an interesting caveat of how contamination was determined. In addition to the contamination described, rare tissues do have very low expression of some

of these contaminating genes. For example, and as seen below, *FGA*, highly expressed in liver, is rarely also lowly expressed elsewhere including the lung and pancreas. Our analysis found it variably expressed in lung, pancreas, small intestine, stomach and whole blood as a result of contamination. While *FGA* is likely variable across all tissues due to contamination, its levels are usually below our mean variance stabilized cut off in our pipeline. So, for tissues with some basal expression (lung and pancreas), we are able to assay *FGA* and observe the relationship to sequencing on a liver day.

We also performed this for *PRSSI* (shown below). *PRSSI* has significantly higher expression (TPM 99,100) than *FGA* (TPM 5,717). Higher expression, leads to higher levels of contamination which were more frequently above our basal expression cut off in our pipeline. For *PRSSI*, small intestine, stomach and spleen all may have possible basal expression. These and 40 other tissues showed variation due to sequencing day contamination.

In the figures for *FGA* and *PRSSI* below, we used $\log_{10}(\text{TPM} + 1)$ data provided from GTEx and showed expression on either liver or pancreas sequencing days respectively for representative tissues. We also show GTEx Portal data of gene expression in \log_{10} sorted by median TPM values. Regardless of possible intrinsic expression in a tissue, overall expression values are right-shifted (higher TPMs) on liver or pancreas sequencing days for these two representative genes.

All of this is done to ultimately say that “possible” expressors still show expression variation as a result of contamination from the dominant tissue source. This contamination is more significant than

basal expression in driving variation.

PRSS1 Expression by Tissue Type

Gene expression for PRSS1 (ENSG00000204983.13)

5. *The paper states that analyses were made with R v3.5.1, but the reporting summary states v3.6.1. No version information for R packages is given. data-table is not a valid R package name (should be data.table).*

Thank you for pointing this out. All of our analyses have now been run in R v3.6.1 and we have revised this in our text accordingly.

6. *Several items used in the paper are listed as n/a in the reporting summary. For example, test statistics (for the linear regression) with confidence intervals, effect sizes, degrees of freedom and p-values, and a full description of statistical parameters and other basic estimates. These should be provided for the linear regression analyses in the paper.*

We have added these parameters, some in the text, others in the new supplemental data for the linear mixed models.

7. *The code used for the analyses should preferably be made available to readers.*

All of our code has now been uploaded to https://github.com/mhalushka/gtex_contamination_code and indicated in the methods section on page 30.

8. *Why were CPM values used for the "second highest tissue expression", rather than TPM values, which would be more comparable to the GTEx TPM values?*

The reviewer is correct and all of the CPM values in Table 1 have now been converted to TPM values.

9. *I think that also regarding expression units (counts, TPMs, VST expression values), the text could be clearer/more precise. For example, it is not clear to me what is meant by "Read counts were scaled using TPM/100 to equalize expression level counts to the other two studies."*

We have gone through our paper and have tried to unify all count types to TPMs or normalization scores where appropriate. Specific to the TPM/100 value, a challenge of single cell RNA-sequencing is the low read counts per cell, which can be as little as 20-50,000. As such, adjusting to a transcripts per million (TPM) level results in an overinflation of an expressed gene's value. This is because many genes that would contribute to a standard sequencing signature when a sample is sequenced at a depth of millions are at 0 based on the low sequencing depth. The TPM values for insulin (*INS*) were generally 100,000-300,000 TPM per cell. If we showed Figure 3e (formerly Figure 2e) with TPM (not TPM/100) values, the Y-axis scale would be at 100,000 not 1,000 and we felt that it looked unreasonable and that we were overstating this point. In the manuscript itself, we wrote a comparison of TPM and TPM/100 (the same as transcripts per 10,000 cells): "From the pancreas sample, insulin (*INS*) represented a highly-expressed, cell type-enriched gene that is exclusive to beta cells (TPM value of 167,144; transcripts per 10,000 cells = 1,671)."

Reviewer #2 (Remarks to the Author):

In this work, the authors analyze RNA-seq data from GTEx tissues and identify signs of potential sample contamination such as observing co-variation clusters of genes highly and specifically enriched in pancreas in 24 unrelated tissues. In a linear model, sample contamination was significantly associated

with a sample of the native tissue expressed on the same day. The authors also analyze genotype differences from DNA and RNA. Also presented are analyses considering contamination implications in eQTL and scRNAseq data. While the study is interesting and quite relevant to the field, I have a few questions as I try to interpret the results, answering/clarifying which could make the study more robust:

1. Fig 1A – what if the genes in the ‘contamination’ cluster are just lowly and variably expressed across individuals in different tissues, or are due to technical artifact such as PCR duplicates instead of widespread cross-sample contamination (especially because the authors describe observing the pancreas genes in as many as 24 tissues)?.

We are confident that we have uncovered an interesting contamination story. Nevertheless, it is also true that occasionally genes are lowly expressed in some tissues (see examples for *FGA* and *PRSSI* above). However, this low expression, cannot explain our findings across tissues. If low expression was the cause, elevated expression in secondary tissues would not correlate with the date of sequencing relative to the true-expressing tissues (Fig. 1b/c). Also, the SNP data seen in Fig. 1e demonstrates the reads in the fibroblast (which is contaminated) were not of the genome of that individual. We uncovered too many of these genomic switches for this to be explained as low and variable expression.

To further show this, we generated this figure showing *KRT4* vs *KRT13* TPM values across different tissues (X and Y axis values are $\log_{10}(\text{TPM}+1)$). Both genes are expressed highly in esophagus and we hypothesize that where they are co-expressed in other tissues, that is the result of similar contamination from esophagus. We see that distinct correlation appearance in liver, fibroblasts, tibia and coronary artery. However, in lung, the genes have a low, non-correlative expression pattern, consistent with basal intrinsic expression.

As described above in reviewer 1’s first question, for reasons unknown to us, the pancreas has the most

highly expressed tissue specific genes of the body. With such high expression, our ability to detect variable expression patterns is increased allowing more opportunities (i.e. tissues) to detect these differences.

A. To directly answer this, one could utilize independent eQTL studies that performing RNA-seq on samples from one tissue, say, subcutaneous adipose eQTL study (10.1016/j.ajhg.2019.09.001, N=434) and run a similar analysis as Fig 1A. This should enable confirming that these samples do not show expression or co-variation of the genes in the ‘contamination’ cluster? A comparison could also be useful for the eQTL analysis in the manuscript.

Thank you for alerting us to this recent adipose sequencing study, which we will refer to as the METSIM study. It further supports how contaminating gene expression exists only when contaminating tissues are sequenced at the same time. Using this new data from METSIM we have generated an additional supplementary figure (5) that plots pairs of contaminating genes from the same tissue source in GTEx and METSIM. In the GTEx adipose samples, the data nicely demonstrates a strong pairwise correlation of contaminating genes from pancreas, esophagus, stomach and skeletal muscle. This indicates the consistency of contamination from one sample to another of these highly expressed contaminating genes. In METSIM adipose, where there was no pancreas, esophagus, stomach or skeletal muscle co-sequenced, there is both a paltry level of reads ascribed to these genes (as noted by $\log_{10}(\text{TPM}+1)$ values) and a clear lack of co-expression of these contamination genes for three of these tissues. Importantly, there is co-expression of skeletal muscle genes *MYBPC1* and *CKM*, which is attributable to direct tissue contamination during adipose harvesting as it is easy to capture muscle during that harvesting step. This is supported by a prior METSIM study (Civelek M et al. Hum Mol Genet 2013) that we had evaluated for overall microRNA expression (Baras AS et al PloS One 2015 and McCall MN et al Genome Res 2017). Here two “myomiRs” miR-133a-3p and miR-1-3p showed overall low, but variable and positive correlation across 100 samples ($R^2=0.39$).

Specific to the eQTL question, our analysis requires read counts and the publicly available data is in TPM. We reached out to the lead author of the study but were unable to obtain the data we needed to investigate these for eQTLs. Additionally, the extremely low levels of counts of these genes (with the exception of the skeletal muscle contamination) would fall below our minimal threshold in our analysis pipeline and would not be considered, rendering this an unfair comparison.

- A. *While the authors compare other datasets in Fig 2d, I'm concerned about a large difference in the number of samples; large deeply sequenced RNA-seq studies are more likely to identify more lowly expressed genes.*

The reviewer is correct that having the deep size of GTEx created the environment for us to detect this form of contamination. There simply are not any other equal-sized studies in which to directly compare our results. We have obtained the next two largest studies, which are clearly not as thorough. However, as noted in Supplementary Data 6 (formerly Supplementary Table 2) we could find evidence of contamination in all paired sample scenarios we investigated.

2. *Table 1 – Why isn't TPM reported for independent study for the 2nd highest tissue for a direct comparison?*

The reviewer is correct and all of the CPM values in Table 1 have now been converted to TPM values.

3. *Fig 1a – there appear to be some sub-clusters in the 'contamination' cluster – do these correspond to different tissues based on the genes highlighted?*

Yes, this overall “contamination” cluster consists of one pituitary gland gene (*PRL*), five clustered pancreas genes (*REG1A*, *GP2*, *PNLIP*, *CELA3A*, and *PRSS1*), one stomach gene (*LIPF*), and lastly two esophagus mucosa genes (*KRT4* and *KRT13*). This overall “contamination cluster” made of genes from multiple different sources is not consistent from tissue to tissue (only 5 of the 9 genes overlap with the transformed fibroblast cluster in Supplementary Figure 1). In general, subclusters with stronger positive correlation are the result of two or more genes from the same tissue source (ex. *KRT4* and *KRT13*) as they would share the same contamination sources.

4. *Fig 1b,c – show that the Z score of the gene PRSS is higher in tissue samples where a pancreas sample was sequenced the same day. It would be reassuring to see violin+jitter plots of TPM of the co-variation cluster genes (PRSS, PNLIP, CLE3A etc) in the 24 tissues where this cluster is observed vs the other 24 where it is not observed as control, classified into if a pancreas sample was sequenced the same day or not.*

Thank you for this suggestion and we have made this into Supplementary Figure 2. As you can see in the figure, for all of the tissues, there is always a higher normalization score for the samples sequenced on the same day as the pancreas, regardless of whether or not the cluster was observed. The reason there are 21 tissues that did not have these clusters is simply because of the lower levels of contamination as seen in the bottom right corner of the figure. These 21 tissues simply did not cross our minimum threshold of mean >5 VST counts.

Normalization score of pancreas contamination

A. be useful to see plots for other genes highly expressed in the pancreas (eg. *INS*) not identified in the co-variation cluster if still there is some trend with the sequencing day.

Below we have posted the plot showing the normalization score of *INS* in samples sequenced the same day as pancreata and those that were not sequenced the same day. Some tissues are missing due to the lack of *INS* expression, which is to be expected as beta cells are a rare cell type of the overall pancreas and median *INS* TPM in pancreas was only 2,326. Despite the low levels, *INS* is always higher on pancreas sequencing days. Using an unadjusted T-test, the resulting p-value is $8.82e-248$ with the mean normalization score in non-pancreas days being -0.60 and on pancreas sequencing days 0.13.

B. Fig 1c – a Wilcoxon Rank Sum Test would aid in quantifying the trend in this plot.

The resulting P-value of the Wilcoxon Rank Sum Test is 5.85×10^{-62} and has been added to the Figure 1 legend.

5. For the allelic incongruency analysis, are the authors accounting for read quality?

Yes, read quality is accounted for. For the example shown in Fig 1e, the average read quality for the base pairs was a Quality Value Score of 37.1, 37.9 for G nucleotides at the locus and 37.2 for the C nucleotides.

6. GTEx 1 fibroblast and GTEx2 esophagus comparison in an interesting example. Were there SNPs in any other genes with detectable expression in fibroblasts that could support contamination? i.e. homozygous genotype in fibroblast but robustly detectable alternate alleles in RNA?

Yes. Other genes supported the contamination. Table 2 reports 5 SNPs in fibroblast cells in two genes (*KRT13* and *KRT4*) that were found supporting this contamination. The limitation in finding more contamination examples was multifactorial. First, the gene has to be expressed at a considerable enough level to be noticeable in the contamination of another tissue. Secondly, there also have to be SNPs in the generally highly conserved protein coding region or untranslated regions. Third, there are gaps in the available GTEx data such that obtaining paired DNA and RNA expression for the same samples was surprisingly limiting.

7. Table 3 mentions read counts – it would be more robust to use TPM here to account for different read depths across samples. Such a table could also be better visualized as a cumulative distribution plot across TPM thresholds.

This was a very good idea and we have replaced the table with a new figure (Figure 2) of the requested cumulative distribution plot across the TPM thresholds. It nicely shows how ~25% of all samples have 0 TPM for these four genes. It also shows how *PRSSI*, which is the most highly expressed of these genes in pancreas (99,100 TPM), has the highest levels of contamination of the four genes

Other minor comments:

Line 91 – is there a citation missing?

The citation has been added.

Lines 97-108 describe an analysis, it would be helpful to the readers if the appropriate figure was referenced

We have added Fig. 1a and Supplementary Fig. 1 comments as appropriate.

Figure 1A and Supplementary Fig 1, it would be more useful to the readers if some genes in other co-variable clusters were also labeled.

We have added descriptions of the alphabetized clusters in the figure legends of Figure 1 and Supplementary Figure 1. In Figure 1, the etiology of cluster A is unknown. Cluster B appears to relate to the percentage of smooth muscle cells in the cluster and Cluster C is a collection of acute phase reactant genes. In Supplementary Figure 1, both clusters A and B are unclear. Cluster C is a collection of cell cycle related genes.

Line 146 – The authors go from co-variation clusters to regression model including isolation date – it would be helpful if the authors clarified in this sentence what was the metric to quantify contamination for each sample.

We have rerun the regression model with a new linear mixed model that more accurately queries the data. All of this is clarified with new descriptions in the manuscript in the methods on pages 25-26 and in the results on pages 8 and 9.

Line 146 – were all tissues considered together in the linear regression model?

We have now focused on only those tissue (N=15 and N=8, respectively) that have >40 samples sequenced on a non-contaminating tissue day (pancreas and esophagus, respectively). This is part of a more robust linear mixed model treating tissues as covariates, which is now seen above and has been added to the manuscript and as Supplementary Data Tables 3 and 4.

Line 103-105 is there a citation?

We have added the citation Muraro MJ et al A Single-Cell Transcriptome Atlas of the Human Pancreas. Cell Syst 2016 which demonstrates acinar-specific expression of these genes (specifically their Table S3).

Line 214 – table reference is wrong

Thank you for catching this. We have replaced the table with a new Figure 2 and have properly referenced it.

Line 293 – figure reference should be 2e?

Correct, thank you for pointing this out. It has been corrected.

Reviewer #3 (Remarks to the Author):

The idea behind this paper is highly innovative, and brings to light an important issue. However, the manuscript lacks sufficient detail on a number of aspects of the study materials and methods that need to be addressed.

1. *One major concern is the lack of information about the sequence runs with goal of more precisely localizing the source of the cross-contamination to a given stage of the sequence run. These comments are included in the comments to the bioRxiv publication, and I think need to be addressed: "Currently the text states that they were all Illumina HiSeq 2000/2500 runs. But the HiSeq 2500 could run in one of two "modes" using either standard "High Output" or "Rapid" chemistry. "Rapid" chemistry could use either "on-board" or cBot clustering. On board clustering trafficked the contents of the run library pool to both lanes of the flow cell through fixed (non-disposable) tubing and this was known to be a source of run-to-run contamination as, to my knowledge, no bleach wash protocol was ever offered by Illumina for the 2500. Whereas high output chemistry pumped lane pools of libraries through disposable tubing while clustering them. This occurred on a separate instrument, called a cBot. For a large project of this sort, one might tacitly presume that the chemistry used would be High Output. But it is also mentioned in the text that dual indexing was used for this project. Dual indexing was not "native" to the 2000/2500 high output chemistry kits -- requiring the addition of custom primer for i7 indexing. If high output (cBot clustered) flow cells were used, then any lane-to-lane or run-to-run contamination detected would likely derive from processing prior to clustering. This would ultimately get at the question as to whether this cross-contamination is the result of index hopping -- a phenomenon largely ascribed to newer Illumina instruments that use patterned flowcell and x-amp clustering chemistry. Another possibility to consider would be whether Illumina's default mismatch=1 indexing was used during demultiplexing and if much of the contamination could be avoided by setting it to mismatch=0."*

We reached out to both GTEx and the Broad Sequencing group for these details as they were not available through standard GTEx metadata elements. Sheila Dodge, General Manager of Broad Genomics, provided information that we included in the discussion (lines 359-363):

Specific to this study, GTEx samples were run on either HiSeq 2000 or 2500 machines in high output mode using cbot clustering. The flowcells were not patterned, and therefore not prone to index-swapping (personal communication, GTEx Help Desk). GTEx's use of custom i7 (later Illumina kit -based) dual indices also reduced the amount of index hopping that can occur.

2. *Additionally, detail should be added to the section at Page 5, line 110-113: (also in the 2nd column of Table 1). They demonstrate that these genes were highly expressed in their native tissues, but how were they identified as variables in unrelated tissues should be discussed. At least, the name of used method should be mentioned. I think this key step ought to be as clear as possible so that other people can reproduce the work.*

Our variation pipeline (ie our method) was described in “Complex Sources of Variation in Tissue Expression Data: Analysis of the GTEx Lung Transcriptome” by Matt McCall et al. *AJHG*, 2016. In response to this request and the request from reviewer 1, all of our code, including this method, has been uploaded to https://github.com/mhalushka/gtex_contamination_code.

On a more minor note,

3. *Fig 1a: Why are these genes grouped in the contamination cluster? Because these genes are not supposed to show up in these tissues?*

The simple answer is that the pancreas genes (*REG1A*, *GP2*, *PNLIP*, *CELA3A*, *PRSS1*) are all contaminating at the same time, based on date of contamination. So, if a sample was sequenced on the same day as a pancreas, it was more likely to be contaminated by all of the pancreas genes. If a sample was not sequenced on the same day as a pancreas tissue, it was likely to have zero sequence reads for any of the pancreas genes. That consistent “variable expression” resulted in the strong correlations among the pancreas genes. The other contaminating genes, *LIPF*, *PRL*, *KRT4* and *KRT13*, also show up in that larger cluster, but that was not a consistent phenomenon across all other tissues. In Supplementary Figure 1, a different set of contaminating genes (*PGC*, *LIPF*) are correlated with the pancreas gene cluster. In other tissues, no other contaminating genes are correlated with the pancreas genes. Invariably, *KRT4* and *KRT13* were always strongly correlated as they also came from the same contaminating source.

4. *Page 8, line 146-149: Since the authors said they used linear regression, why not list what equation they really used. What was the dependent variable in the model? Was the nucleic acid isolation date the only independent variable? Is there any covariate? What was the p-value and estimate for intercept term? There is more information needed here to support the conclusion. Additionally, it would be better to use model comparison instead of p-values from linear regression for this purpose.*

In response to all reviewers comments, we have generated a new, more robust linear mixed models (above). We have updated the paper and generated 4 new Supplementary Data tables.. The models are now run independently for 4 pancreas and 2 esophagus contaminating genes across all GTEx tissues. These models show overwhelming support for pancreas and esophagus sequencing day as the cause of

elevated expression levels of highly-expressed pancreas or esophagus genes in non-pancreas and esophagus tissues. A new, extensive write up of these models are now incorporated into the method on pages 25-26 and in the results on pages 8-9.

5. *The p-value is 1.21e-67 in Fig 1b (Page 7, line 123), but 2.66e-75 in Page 8, line 154. On a related note, why not conduct t-test between these two groups?*

Thank you for the suggestion. We have run that as a t-test and the results are:

Mean in group sequenced on pancreas day: 0.1855

Mean in group not sequenced on pancreas day: -0.0697

95 CI: -0.9743 to -0.7907

t = -18.888, df = 528.52, p.val: 3.430e-61

6. *The meaning of y-axis in Fig. 2c is not clear.*

For the y-axis in Fig 2c (Now Fig 3c) we have added “N=” indicating these numbers are the amount of samples included in the graphic and explained this in the figure legend. We have also changed the colors of Fig 3a and Fig 3e to hopefully reduce any other confusion across the different studies used and clearly indicate the colors of Fig 3c and Fig 3d are of the same study.

7. *Typos: Page 3, line 64: Limitations exists should be exist.*

Thank you, this has been corrected.

We hope these revisions satisfy the reviewers concerns and we look forward to this paper reaching a wide audience.

REVIEWERS' COMMENTS:

Reviewer #1 (Remarks to the Author):

I would like to thank the authors for performing additional checks and analyses in order to address my initial comments. I think it would be useful to also briefly mention these results in the revised manuscript, to avoid other readers worrying about the same issues.

In addition, I think the description of the different abundance units is still somewhat confusing in some places:

- Instead of TPM/100, I would suggest using TP10K, which is frequently used in the single-cell literature.
- On line 178, it is not clear what TPM threshold was set at the VST step. From the earlier description, it seems to be a threshold on the VST expression values.
- on lines 560-561, it is not clear what it means to "scale read counts using TPM/100". Were the raw read counts scaled (and in what way), or were rather the TPM values (divided by 100) used (these are not read counts, and should not be referred to as such)?

Typos and minor comments:

- line 160: "same day a" -> "same day as a"
- line 168: "significance of with"
- fig. 2: this is not typically referred to as a density plot, but a cumulative distribution plot
- line 300: should not be transcripts per 10,000 _cells_
- line 309: what does "adjustment by" 6 or 10 PCs mean? From the methods, it seems that the tSNE was calculated from the first 6 or 10 PCs.
- line 327: missing parenthesis after respectively
- several places: the "O" in GEO stands for Omnibus; thus, "GEO" is enough (not "GEO Omnibus")
- line 463: the "median absolute deviation" typically refers to the median of the absolute deviations from the _median_, not the mean. Please clarify what \bar{x} stands for, and over which samples it was calculated.
- line 504: unmatched parenthesis
- line 514: missing date when the website was visited
- line 517: what kind of scaling was used?
- line 552: the division of TPMs by 10 is already included in the formula on the previous line. Were they divided again?
- line 553: how does division of the TPMs by 10 "account for the effect of 0s in the downstream analysis"? Just dividing the values will not change whether a value is zero or not.

Reviewer #2 (Remarks to the Author):

Nieuwenhuis et al. have updated their manuscript to appropriately address reviewer comments and the

current version describes and presents more robust analyses more clearly. The figure updates and method clarifications are helpful. It is reassuring to see the new supplementary figure 3 - that another large, deeply sequenced RNA-seq study on only Adipose tissue did not pick up patterns of contamination as can be seen in GTEx Adipose RNA-seq data. Several plots in the reviewer response document are also useful to see. I also appreciate the authors sharing their code. I think all updates have resulted in a stronger manuscript.

Reviewer #3 (Remarks to the Author):

The authors did an excellent job addressing prior concerns.

REVIEWERS' COMMENTS:

Reviewer #1 (Remarks to the Author):

I would like to thank the authors for performing additional checks and analyses in order to address my initial comments. I think it would be useful to also briefly mention these results in the revised manuscript, to avoid other readers worrying about the same issues.

We have added comments to the manuscript touching on the responses we provided in the initial reviewer response that we had not added to the manuscript at that time. Additionally, because we have made this an open review, all of those materials should also be available to readers who would like to dig further into those same concerns through the use of our reviewer response.

In addition, I think the description of the different abundance units is still somewhat confusing in some places:

- Instead of TPM/100, I would suggest using TP10K, which is frequently used in the single-cell literature.

We have changed all instances of TPM/100 to TP10K.

- On line 178, it is not clear what TPM threshold was set at the VST step. From the earlier description, it seems to be a threshold on the VST expression values.

We have now clarified this sentence to read: For all tissues, median normalized scores were higher on pancreas sequencing dates, and the division between the 26 tissues with clustering vs. the 21 without was a result of the high threshold for VST normalized counts we set for our pipeline.

- on lines 560-561, it is not clear what it means to "scale read counts using TPM/100". Were the raw read counts scaled (and in what way), or were rather the TPM values (divided by 100) used (these are not read counts, and should not be referred to as such)?

This was clarified to read: This was done to create TP10K values to account for single cell read depth and the addition of 1 was to limit $TPM_{i,j}$ was then divided by 10, to be on the order of 100,000 transcripts to account for single cell read depth and the effect of 0s in downstream analysis.

Typos and minor comments:

Thank you for catching all of these small problems. They have all been corrected.

- line 160: "same day a" -> "same day as a"

- line 168: "significance of with"

- fig. 2: this is not typically referred to as a density plot, but a cumulative distribution plot

- line 300: should not be transcripts per 10,000 _cells_

- line 309: what does "adjustment by" 6 or 10 PCs mean? From the methods, it seems that the tSNE was

calculated from the first 6 or 10 PCs.

- line 327: missing parenthesis after respectively

- several places: the "O" in GEO stands for Omnibus; thus, "GEO" is enough (not "GEO Omnibus")

- line 463: the "median absolute deviation" typically refers to the median of the absolute deviations from the `_median_`, not the mean. Please clarify what `x_bar` stands for, and over which samples it was calculated.

- line 504: unmatched parenthesis

- line 514: missing date when the website was visited

- line 517: what kind of scaling was used?

- line 552: the division of TPMs by 10 is already included in the formula on the previous line. Were they divided again?

- line 553: how does division of the TPMs by 10 "account for the effect of 0s in the downstream analysis"? Just dividing the values will not change whether a value is zero or not.

Reviewer #2 (Remarks to the Author):

Nieuwenhuis et al. have updated their manuscript to appropriately address reviewer comments and the current version describes and presents more robust analyses more clearly. The figure updates and method clarifications are helpful. It is reassuring to see the new supplementary figure 3 - that another large, deeply sequenced RNA-seq study on only Adipose tissue did not pick up patterns of contamination as can be seen in GTEx Adipose RNA-seq data. Several plots in the reviewer response document are also useful to see. I also appreciate the authors sharing their code. I think all updates have resulted in a stronger manuscript.

Thank you.

Reviewer #3 (Remarks to the Author):

The authors did an excellent job addressing prior concerns.

Thank you.